# STEP-DPO: STEP-WISE PREFERENCE OPTIMIZATION FOR LONG-CHAIN REASONING OF LLMS

## ABSTRACT

Mathematical reasoning presents a significant challenge for Large Language Models (LLMs) due to the extensive and precise chain of reasoning required for accuracy. Ensuring the correctness of each reasoning step is critical. To address this, we aim to enhance the robustness and factuality of LLMs by learning from human feedback. However, Direct Preference Optimization (DPO) has shown limited benefits for long-chain mathematical reasoning, as models employing DPO struggle to identify detailed errors in incorrect answers. This limitation stems from a lack of fine-grained process supervision. We propose a simple, effective, and data-efficient method called Step-DPO, which treats individual reasoning steps as units for preference optimization rather than evaluating answers holistically. Additionally, we have developed a data construction pipeline for Step-DPO, enabling the creation of a high-quality dataset containing 10K step-wise preference pairs. We also observe that in DPO, the data generated by the policy model is more effective than that produced by humans or GPT-4, due to the former's in-distribution nature. Our findings demonstrate that as few as 10K preference data pairs and fewer than 500 Step-DPO training steps can yield a nearly 3% gain in accuracy on MATH for models with over 70B parameters. Notably, Step-DPO, when applied to Qwen2-72B-Instruct, achieves scores of 70.8% and 94.0% on the test sets of MATH and GSM8K, respectively, surpassing a series of closed-source models, including GPT-4-1106, Claude-3-Opus, and Gemini-1.5-Pro.

## 1 INTRODUCTION

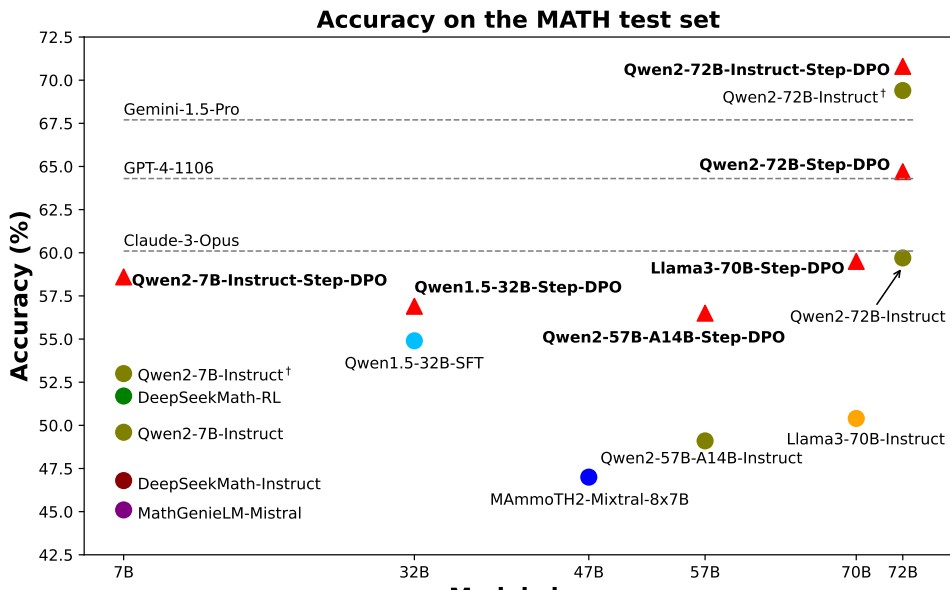

Figure 1: Accuracy on the MATH test set across models fine-tuned by Step-DPO and other state-of-the-art models. [†]: reproduced result using our prompt.

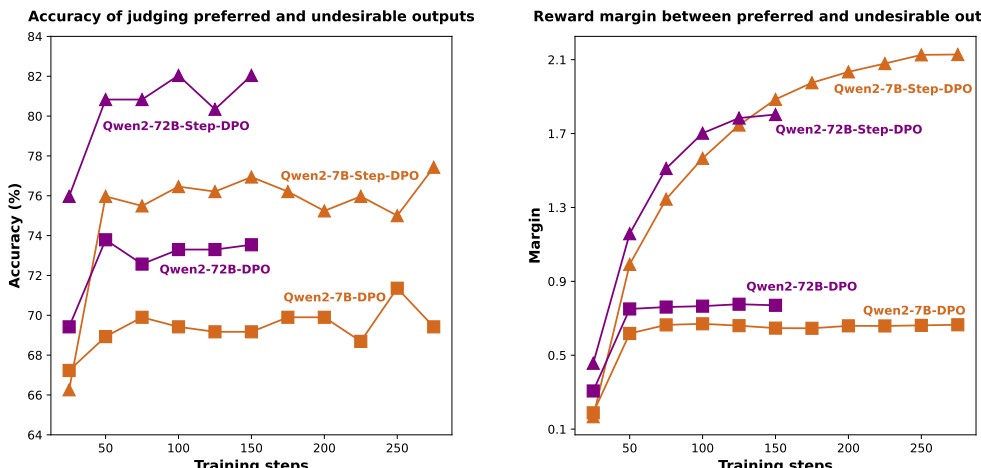

Figure 2: **Left**: Accuracy of judging preferred or undesirable outputs on the validation set during training. **Right**: Reward margins between preferred and undesirable outputs on the validation set during training. More details about these experiments are given in Appendix A.3.

Mathematical reasoning is recognized as a critical long-chain reasoning ability in Large Language Models (LLMs). This task is particularly challenging due to the often extensive chain of thought required, which can include numerous reasoning steps. Any error in these steps can lead to an incorrect final answer.

Numerous studies (Yu et al., 2023; Luo et al., 2023; Yue et al., 2023; Liu & Yao, 2024; Lu et al., 2024; Li et al., 2024; Shao et al., 2024; Xin et al., 2024; Yue et al., 2024; Tang et al., 2024) have proposed various data augmentation techniques during the supervised fine-tuning (SFT) stage to enhance alignment. However, models in the SFT process are prone to hallucinations, resulting in saturated performance. A potential reason for this, as highlighted in Hong et al. (2024), is that as the probability of preferred outputs increases, so does the probability of undesirable ones. This phenomenon makes the model more likely to make errors in long-chain reasoning. Therefore, it is essential to develop methods to suppress the likelihood of undesirable outputs.

Recently, Direct Preference Optimization (DPO) (Rafailov et al., 2024) has been proposed for alignment using pair-wise preference data and is popular due to its simplicity. Despite its effectiveness in chat benchmarks (Tunstall et al., 2023; Zheng et al., 2024), DPO offers minimal benefits for long-chain mathematical reasoning. As shown in Fig. 2 (left), models using vanilla DPO perform poorly in distinguishing between preferred and undesirable outputs, failing to identify errors in rejected answers. Additionally, Fig. 2 (right) shows that the reward margin (i.e., the gap between the rewards of preferred and undesirable outputs) is limited for models using vanilla DPO and plateaus with further training. These findings indicate that models fine-tuned with vanilla DPO cannot pinpoint detailed errors in incorrect answers, hindering the improvement of reasoning abilities.

In this work, we introduce Step-DPO, where each intermediate reasoning step is treated as the basic unit for preference optimization. As illustrated in Fig. 3, unlike vanilla DPO, which only considers preference optimization between complete answers (i.e., $p(y_{win}|x)$ and $p(y_{lose}|x)$), Step-DPO examines the step-by-step answer (i.e., $y = [s_1, ..., s_n]$) and specifically targets the first erroneous reasoning step. Step-DPO aims to select a correct reasoning step and reject an incorrect one, given a math problem and several initial correct reasoning steps (i.e., maximize $p(s_{win}|x; s_1, s_2, ..., s_{k-1})$ and minimize $p(s_{lose}|x; s_1, s_2, ..., s_{k-1})$). This transition allows the model to easily locate erroneous tokens for effective optimization, significantly enhancing long-chain reasoning.

Moreover, we present an effective and economical pipeline to collect pair-wise preference data, resulting in a high-quality dataset for Step-DPO. This dataset contains approximately 10K samples, each consisting of: 1) a mathematical problem, 2) prior reasoning steps, 3) the chosen step, and 4) the rejected step. Our three-step pipeline for dataset construction includes: 1) Error collection, 2) Step localization, and 3) Rectification. Notably, the chosen reasoning step is generated by the policy

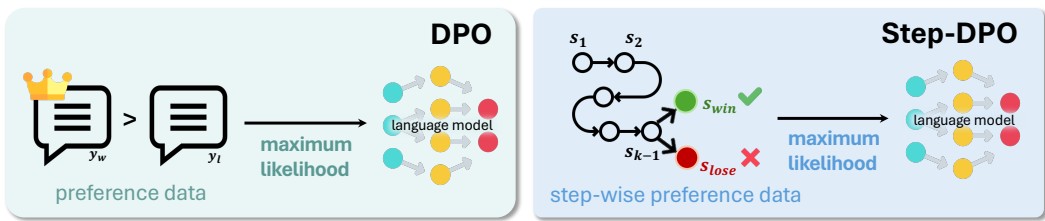

Figure 3: Comparison between DPO and Step-DPO.

model[1] itself, as we find that in-distribution data (i.e., self-generated data) is more effective than out-of-distribution data (e.g., data written by humans or GPT-4) for Step-DPO, as shown in Table 4.

With this curated dataset, mathematical reasoning performance can be significantly boosted with only hundreds of training steps, as demonstrated in Fig. 1. For instance, fine-tuning Qwen-72B-Instruct with Step-DPO results in a model achieving 70.8% accuracy on MATH and 94.0% on GSM8K, surpassing a series of closed-source models, including GPT-4-1106, Claude-3-Opus, and Gemini-1.5-Pro.

## 2 RELATED WORKS

### 2.1 MATHEMATICAL REASONING

Large Language Models (LLMs) have exhibited substantial reasoning capabilities, primarily due to their auto-regressive nature, which allows them to predict the next token based on contextual information. However, these models still struggle with long-chain reasoning tasks, particularly in mathematical contexts. Several prior studies (Yao et al., 2024; Chen et al., 2024; Yoran et al., 2023; Li et al., 2023; Tong et al., 2024; Fu et al., 2022; Zhou et al., 2022) have attempted to enhance the Chain-of-Thought (CoT) inference framework (Wei et al., 2022) to address this issue. While these efforts have led to significant improvements in certain tasks, they have not fully mitigated common hallucinations and have limited generalizability across all reasoning tasks.

Another research direction (Yu et al., 2023; Luo et al., 2023; Yue et al., 2023; Liu & Yao, 2024; Lu et al., 2024; Xu et al., 2024; Li et al., 2024; Shao et al., 2024; Xin et al., 2024; Zhou et al., 2024; Liu et al., 2023; Ying et al., 2024; Yue et al., 2024; Tang et al., 2024; Mitra et al., 2024; Yuan et al., 2023) focuses on various data augmentation techniques, such as rephrasing, extension, and evolution, for supervised fine-tuning (SFT). These methods have significantly enhanced the reasoning abilities of LLMs, but their performance plateaus once the data reaches a certain volume. Additionally, methods like those proposed by Wang et al. (2023a); Liao et al. (2024); Toshniwal et al. (2024); Gou et al. (2023) employ external tools, such as Python, to substantially reduce calculation errors.

Other approaches (Azerbayev et al., 2023; Shao et al., 2024; Lin et al., 2024; Ying et al., 2024; Wang et al., 2023c) involve continued pre-training on extensive, high-quality math-related datasets, which markedly improve mathematical reasoning capabilities. Recent studies (Xu et al., 2024; Ying et al., 2024) have explored reinforcement learning to mitigate hallucinations in mathematical reasoning. Works like Lightman et al. (2023); Shao et al. (2024); Wang et al. (2023b); Jiao et al. (2024); Hwang et al. (2024) emphasize the importance of step-by-step verification in reinforcement learning for mathematical problems. However, these methods still rely on the quality of the reward model and require the complex training pipelines of RLHF. Building on this line of research, we propose Step-DPO, a simpler, more effective, and more efficient method.

### 2.2 REINFORCEMENT LEARNING FROM HUMAN FEEDBACK

Supervised fine-tuning (SFT) can align models with human preferences. However, as the probability of preferred outputs increases, so does the likelihood of undesirable ones, leading to hallucinations. To generate more reliable outputs, Reinforcement Learning from Human Feedback (RLHF) (Christiano et al., 2017; Ouyang et al., 2022) has been introduced for LLM alignment. This approach involves

---

[1]The policy model refers to the model that we are optimizing, and it is usually initialized with the SFT model.

training a reward model with comparison data and then using this reward model to optimize the policy model. The final performance heavily depends on the quality of the reward model, and the training pipeline is quite complex.

To simplify this process, Direct Preference Optimization (DPO) (Rafailov et al., 2024) was proposed, which directly uses pair-wise preference data for model optimization. This transition significantly streamlines the training pipeline. While DPO has proven effective in chat benchmarks, it offers only marginal benefits for mathematical reasoning. Inheriting the principles of DPO, Step-DPO is specifically designed for long-chain reasoning and has shown significant performance improvements in solving math word problems.

## 3  STEP-DPO

In this section, we elaborate on the proposed Step-DPO. First, we present step-wise formulation in Sec. 3.1, a novel approach designed to enhance long-chain reasoning abilities by building on DPO. Next, in Sec. 3.2, we illustrate a pipeline for constructing the step-wise preference dataset for Step-DPO. Both components are essential for achieving the desired performance improvements.

### 3.1  STEP-WISE FORMULATION

**Preliminary.**  Reinforcement Learning from Human Feedback (RLHF) (Christiano et al., 2017) is an effective approach for enhancing the robustness, factuality, and safety of LLMs (Ouyang et al., 2022). RLHF consists of two training phases: 1) reward model training, and 2) policy model training. However, the final performance of RLHF is highly sensitive to various hyperparameters in both phases, necessitating meticulous tuning.

To avoid this complex training pipeline, Rafailov et al. (2024) proposed Direct Preference Optimization (DPO), which directly uses pair-wise preference data to optimize the policy model with an equivalent optimization objective. Specifically, given an input prompt $x$, and a preference data pair $(y_{win}, y_{lose})$, DPO aims to maximize the probability of the preferred output $y_{win}$ and minimize that of the undesirable output $y_{lose}$. The optimization objective is formulated as:

$$\mathcal{L}_{DPO}(\theta) = -\mathbb{E}_{(x,y_{win},y_{lose})\sim D}[\log \sigma(\beta \log \frac{\pi_\theta(y_{win}|x)}{\pi_{ref}(y_{win}|x)} - \beta \log \frac{\pi_\theta(y_{lose}|x)}{\pi_{ref}(y_{lose}|x)})], \quad (1)$$

where $D$ is the pair-wise preference dataset, $\sigma$ is the sigmoid function, $\pi_\theta(\cdot|x)$ is the policy model to be optimized, $\pi_{ref}(\cdot|x)$ is the reference model kept unchanged during training, and the hyperparameter $\beta$ controls the distance from the reference model.

**Our Solution.**  While DPO has proven effective in chat benchmarks, it brings only marginal improvements for long-chain reasoning tasks such as mathematical problems, as shown in Fig. 2 and Table 3. This limitation arises because most undesirable answers in these tasks may not contain errors initially; the first error often appears midway through the reasoning process. Rejecting an entire undesirable answer in DPO may also discard preceding correct reasoning steps, introducing significant noise and negatively impacting training.

Analogous to how teachers correct students by pinpointing specific errors rather than dismissing entire answers, our proposed Step-DPO provides more detailed supervision by identifying the specific erroneous reasoning step. This granular focus allows the model to swiftly locate, rectify, and further avoid erroneous steps.

Specifically, the answer $y$ can be decomposed into a sequence of reasoning steps $y = [s_1, \ldots, s_n]$, where $s_i$ is the $i$-th reasoning step. As illustrated in Fig. 3, given a prompt $x$ and a series of initial correct reasoning steps $s_{1\sim k-1} = [s_1, \ldots, s_{k-1}]$, Step-DPO aims to maximize the probability of the correct next reasoning step $s_{win}$ and minimize the probability of the incorrect one $s_{lose}$. This objective can be formulated as:

$$\mathcal{L}(\theta) = -\mathbb{E}_{(x,s_{1\sim k-1},s_{win},s_{lose})\sim D}[\log \sigma(\beta \log \frac{\pi_\theta(s_{win}|x; s_{1\sim k-1})}{\pi_{ref}(s_{win}|x; s_{1\sim k-1})} - \beta \log \frac{\pi_\theta(s_{lose}|x; s_{1\sim k-1})}{\pi_{ref}(s_{lose}|x; s_{1\sim k-1})})].$$

$$(2)$$

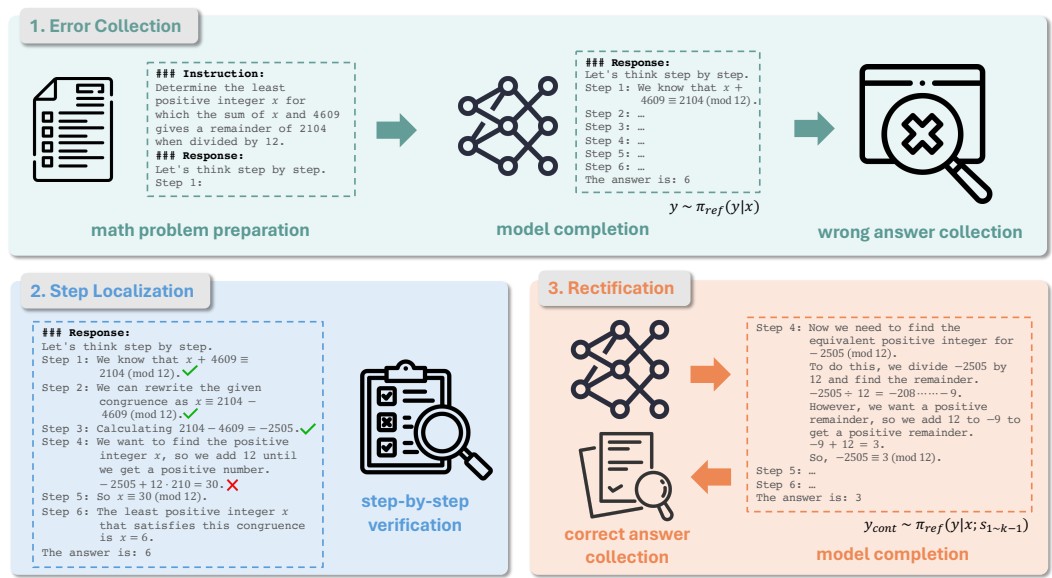

Figure 4: Data construction pipeline for Step-DPO.

---

**prompt**:

    Determine the least positive integer $x$ for which the sum of $x$ and 4609 gives a remainder of 2104 when divided by 12.

**initial_reasoning_steps**:

    Let's think step by step.

    Step 1: We know that $x + 4609 \equiv 2104 \pmod{12}$.

    Step 2: We can rewrite the given congruence as $x \equiv 2104 - 4609 \pmod{12}$.

    Step 3: Calculating $2104 - 4609 = -2505$.

    Step 4:

**chosen_step**:

    Now we need to find the equivalent positive integer for $-2505 \pmod{12}$.

    To do this, we divide -2505 by 12 and find the remainder. $-2505 \div 12 = -208$ remainder $-9$.

    However, we want a positive remainder, so we add 12 to -9 to get a positive remainder. $-9+12=3$.

    So, $-2505 \equiv 3 \pmod{12}$.

**rejected_step**:

    We want to find the positive integer $x$, so we add 12until we get a positive number. $-2505 + 12 \cdot 210 = 30$.

---

Figure 5: An example of preference data sample for Step-DPO.

## 3.2 IN-DISTRIBUTION TRAINING DATA CONSTRUCTION

According to the optimization target of Step-DPO, we need to create a corresponding high-quality pair-wise preference dataset. Each data sample should comprise four entries: 1) prompt $x$; 2) initial reasoning steps $s_{1 \sim k-1} = [s_1, \ldots, s_{k-1}]$; 3) preferred reasoning step $s_{win}$; 4) undesirable reasoning step $s_{lose}$, as shown in Fig. 5. To obtain a high-quality dataset, we propose a data construction pipeline illustrated in Fig. 4, which includes the following three steps.

**Error collection.** First, we collect a set $D_0 = \{(x, \hat{y})\}$ of mathematical problems $x$ with ground-truth answers $\hat{y}$. Each mathematical problem $x$ is then used as a prompt to infer answers using the initial model $\pi_{ref}$. Before inference, we add the step-wise Chain-of-Thought (CoT) prefix for prompting, i.e., `"Let's think step by step. Step 1:"`. This ensures that the model's inference results are structured into multiple reasoning steps, with each step explicitly starting with `"Step i:"`. Upon completion of inference, we obtain the model answers $y$ for each mathematical problem $x$. We then select instances where the final answer $y$ differs from the ground truth $\hat{y}$, resulting in a dataset of erroneous inference results, denoted as $D_1 = \{(x, \hat{y}, y) | x \in D_0\}$.

**Step localization.** Given that each erroneous inference result is explicitly presented as a sequence of reasoning steps $y = [s_1, s_2, \ldots, s_n]$, we proceed to verify the correctness of each reasoning step until we find the first error and record its step number $k$. This process can be done manually or using GPT-4. We select $s_k$ as the erroneous reasoning step $s_{lose}$, resulting in a dataset that contains the erroneous steps, denoted as $D_2 = \{(x, \hat{y}, s_{1 \sim k-1}, s_{lose})|x \in D_1\}$.

**Rectification.** To obtain the corresponding correct reasoning step for each sample in $D_2$, we need to sample multiple outputs $y_{cont}$ by inferring the model $\pi_{ref}$ with the prompt $x$ and the preceding correct reasoning steps $s_{1 \sim k-1}$. This process is formulated as:

$$y_{cont} \sim \pi_{ref}(y|x; s_{1 \sim k-1}). \tag{3}$$

We retain those outputs where the final answer matches the ground truth. Among the remaining outputs, we select the first reasoning step in $y_{cont}$ as $s_{win}$, resulting in the final dataset $D = \{(x, s_{1 \sim k-1}, s_{lose}, s_{win})|x \in D_2\}$. An example of a resulting data sample is shown in Fig. 5.

Notably, some cases may have correct final answers but erroneous intermediate reasoning steps. Therefore, we may need to further filter out samples where $s_{win}$ is incorrect, which can be done manually or by GPT-4. We omit this process in the notations for simplicity, and more details are provided in Appendix A.1.

It is important to note that the data pipeline is user-friendly. In this data pipeline, humans or GPT-4 are only required to locate errors and verify the correctness of reasoning steps, and they do not need to write answers or rectifications by themselves.

We also note that the use of in-distribution data is crucial. When selecting $s_{win}$, we use outputs generated by the model $\pi_{ref}$ rather than answers rectified by humans or GPT-4. Since human or GPT-4 rectified answers $s_{win}^{ood}$ are out-of-distribution (OOD) regarding the model $\pi_{ref}$, the log-probability of outputting $s_{win}^{ood}$ (i.e., $\log \pi_{ref}(s_{win}^{ood}|x)$) is significantly lower than that of an in-distribution (ID) output $\log \pi_{ref}(s_{win}^{id}|x)$. Moreover, it is challenging for the policy model $\pi_\theta$ to learn to increase the probability of $s_{win}^{ood}$ due to gradient decay issues (detailed in Appendix A.4). Consequently, adopting self-generated in-distribution data as the preferred answer proves to be a more effective way of aligning with human preferences.

## 4 EXPERIMENTS

In this section, we first introduce the experimental setup in Sec. 4.1. Then, we present the main results in Sec. 4.2, which include an exhaustive performance comparison. Moreover, we conduct an extensive ablation study in Sec. 4.3. Finally, a few demonstrations are shown in Sec. 4.4 to further understand Step-DPO.

### 4.1 EXPERIMENTAL SETUP

**Network Architecture.** Our experiments are based on various base models, including the Qwen2 and Qwen1.5 series (Bai et al., 2023), Meta-Llama-3-70B (Touvron et al., 2023), and deepseek-math-7b-base (Shao et al., 2024).

**Datasets.** In supervised fine-tuning (SFT), we use augmented mathematical problems from Meta-Math (Yu et al., 2023) and MMIQC (Liu & Yao, 2024) to infer step-by-step responses with DeepSeek-Math, as the SFT data used in DeepSeekMath (Shao et al., 2024) is not publicly available. After filtering out responses with erroneous final answers, we obtain 374K SFT data. Of these, 299K are used for SFT, and the remainder is used for further Step-DPO training.

In the Step-DPO phase, alongside the remaining SFT data, we also incorporate a subset of AQuA (Ling et al., 2017). These datasets are processed as described in Sec. 3.2, resulting in 10K pair-wise preference data for Step-DPO.

For evaluation, we use the widely adopted MATH (Hendrycks et al., 2021) and GSM8K (Cobbe et al., 2021) datasets. Accuracy in these datasets serves as the evaluation metric. The MATH test set contains 5000 mathematical problems spanning 5 difficulty levels and 7 subjects, including

Table 1: Math reasoning performance comparison on MATH and GSM8K across various models. general: general-purpose model. open: open-source.

| Model | size | general | open | MATH (%) | GSM8K (%) |
|---|---|---|---|---|---|
| GPT-3.5-Turbo | - | ✓ | ✗ | 42.5 | 92.0 |
| Gemini-1.5-Pro (Feb) (Reid et al., 2024) | - | ✓ | ✗ | 58.5 | 91.7 |
| Gemini-1.5-Pro (May) (Reid et al., 2024) | - | ✓ | ✗ | 67.7 | 90.8 |
| Claude-3-Opus | - | ✓ | ✗ | 60.1 | 95.0 |
| GPT-4-1106 (Achiam et al., 2023) | - | ✓ | ✗ | 64.3 | 91.4 |
| GPT-4-Turbo-0409 (Achiam et al., 2023) | - | ✓ | ✗ | 73.4 | 93.7 |
| GPT-4o-0513 | - | ✓ | ✗ | 76.6 | 95.8 |
| Llama-3-8B-Instruct (Touvron et al., 2023) | 8B | ✓ | ✓ | 30.0 | 79.6 |
| Qwen2-7B-Instruct (Bai et al., 2023) | 7B | ✓ | ✓ | 49.6 | 82.3 |
| Llama-3-70B-Instruct (Touvron et al., 2023) | 70B | ✓ | ✓ | 50.4 | 93.0 |
| DeepSeek-Coder-V2-Instruct (Zhu et al., 2024) | 236B | ✗ | ✓ | 75.7 | 94.9 |
| Code-Llama-7B (Roziere et al., 2023) | 7B | ✗ | ✓ | 13.0 | 25.2 |
| MAmooTH-CoT (Yue et al., 2023) | 7B | ✗ | ✓ | 10.4 | 50.5 |
| WizardMath (Luo et al., 2023) | 7B | ✗ | ✓ | 10.7 | 54.9 |
| MetaMath (Yu et al., 2023) | 7B | ✗ | ✓ | 19.8 | 66.5 |
| MetaMath-Mistral-7B (Yu et al., 2023) | 7B | ✗ | ✓ | 28.2 | 77.7 |
| MathScale-Mistral Tang et al. (2024) | 7B | ✗ | ✓ | 35.2 | 74.8 |
| InternLM-Math-7B (Ying et al., 2024) | 7B | ✗ | ✓ | 34.6 | 78.1 |
| Xwin-Math-Mistral-7B (Li et al., 2024) | 7B | ✗ | ✓ | 43.7 | 89.2 |
| MAmmoTH2-7B-Plus (Yue et al., 2024) | 7B | ✗ | ✓ | 45.0 | 84.7 |
| MathGenieLM-Mistral (Lu et al., 2024) | 7B | ✗ | ✓ | 45.1 | 80.5 |
| InternLM-Math-20B (Ying et al., 2024) | 20B | ✗ | ✓ | 37.7 | 82.6 |
| MathGenieLM-InternLM2 (Lu et al., 2024) | 20B | ✗ | ✓ | 55.7 | 87.7 |
| DeepSeekMath-Instruct (Shao et al., 2024) | 7B | ✗ | ✓ | 46.8 | 82.9 |
| DeepSeekMath-RL (Shao et al., 2024) | 7B | ✗ | ✓ | 51.7 | 88.2 |
| DeepSeekMath-RL + Step-DPO | 7B | ✗ | ✓ | 53.2 (+1.5) | 88.7 (+0.5) |
| Qwen2-7B-Instruct (Bai et al., 2023) | 7B | ✓ | ✓ | 49.6 | 82.3 |
| Qwen2-7B-Instruct‡ | 7B | ✓ | ✓ | 53.0 | 85.5 |
| Qwen2-7B-Instruct + Step-DPO | 7B | ✓ | ✓ | 58.6 (+5.6) | 87.9 (+2.4) |
| Qwen2-7B-SFT† | 7B | ✗ | ✓ | 54.8 | 88.2 |
| Qwen2-7B-SFT + Step-DPO | 7B | ✗ | ✓ | 55.8 (+1.0) | 88.5 (+0.3) |
| Qwen1.5-32B-SFT† | 32B | ✗ | ✓ | 54.9 | 90.0 |
| Qwen1.5-32B-SFT + Step-DPO | 32B | ✗ | ✓ | 56.9 (+2.0) | 90.9 (+0.9) |
| Qwen2-57B-A14B-SFT† | 57B | ✗ | ✓ | 54.6 | 89.8 |
| Qwen2-57B-A14B-SFT + Step-DPO | 57B | ✗ | ✓ | 56.5 (+1.9) | 90.0 (+0.2) |
| Llama-3-70B-SFT† | 70B | ✗ | ✓ | 56.9 | 92.2 |
| Llama-3-70B-SFT + Step-DPO | 70B | ✗ | ✓ | 59.5 (+2.6) | 93.3 (+1.1) |
| Qwen2-72B-SFT† | 72B | ✗ | ✓ | 61.7 | 92.9 |
| Qwen2-72B-SFT + Step-DPO | 72B | ✗ | ✓ | 64.7 (+3.0) | 93.9 (+1.0) |
| Qwen2-72B-Instruct (Bai et al., 2023) | 72B | ✓ | ✓ | 59.7 | 91.1 |
| Qwen2-72B-Instruct ‡ | 72B | ✓ | ✓ | 69.4 | 92.4 |
| Qwen2-72B-Instruct + Step-DPO ‡ | 72B | ✓ | ✓ | **70.8** (+1.4) | **94.0** (+1.6) |

[†] Supervised fine-tuned models with our 299K SFT data based on the open-source base model.
[‡] Reproduced using our prompt

Table 2: Math reasoning performance comparison on compitition-level math problems, i.e., AIME 2024 and Odyssey-MATH. Note that the training data for Step-DPO is the same as before.

| Model | size | open | AIME | Odyssey-MATH (%) |
|-------|------|------|------|------------------|
| Gemini-1.5-Pro (Reid et al., 2024) | - | ✗ | 2 / 30 | 45.0 |
| Claude-3-Opus | - | ✗ | 2 / 30 | 40.6 |
| GPT-4-1106 (Achiam et al., 2023) | - | ✗ | 1 / 30 | 49.1 |
| GPT-4-Turbo-0409 (Achiam et al., 2023) | - | ✗ | 3 / 30 | 46.8 |
| GPT-4o-0513 | - | ✗ | 2 / 30 | 53.2 |
| DeepSeek-Coder-V2-Lite-Instruct (Zhu et al., 2024) | 16B | ✓ | 0 / 30 | 44.4 |
| Llama-3-70B-Instruct (Touvron et al., 2023) | 70B | ✓ | 1 / 30 | 27.9 |
| DeepSeek-Coder-V2-Instruct (Zhu et al., 2024) | 236B | ✓ | 4 / 30 | 53.7 |
| Qwen2-72B-SFT[†] | 72B | ✓ | 1 / 30 | 44.2 |
| Qwen2-72B-SFT + Step-DPO | 72B | ✓ | 3 / 30 | 47.0 (+2.8) |
| Qwen2-72B-Instruct (Bai et al., 2023) | 72B | ✓ | 5 / 30 | 47.0 |
| Qwen2-72B-Instruct + Step-DPO | 72B | ✓ | 4 / 30 | 50.1 (+3.1) |

[†] Supervised fine-tuned models with our 299K SFT data based on the open-source base model.

Table 3: Performance comparison between DPO and Step-DPO. We use only 5K data for training in this ablation study.

| Model | Qwen2-7B-SFT | Qwen2-7B-SFT + DPO (5K) | Qwen2-7B-SFT + Step-DPO (5K) |
|-------|--------------|-------------------------|------------------------------|
| **MATH** (%) | 54.8 | 55.0 | **55.8** |
| Model | Qwen2-72B-SFT | Qwen2-72B-SFT + DPO (5K) | Qwen2-72B-SFT + Step-DPO (5K) |
| **MATH** (%) | 61.7 | 62.5 | **64.1** |

algebra, counting and probability, geometry, intermediate algebra, number theory, prealgebra, and precalculus. The GSM8K test set includes 1319 mathematical problems, each with a step-by-step solution and a ground-truth answer. The problems in GSM8K are generally easier than those in MATH. Besides, we also use completition-level problems in American Invitational Mathematics Examination (AIME) (MAA, 2024) and Odyssey-MATH (Netmind.AI, 2024) to evaluate the math reasoning capabilities in solving hard problems.

**Implementation Details.** First, we use the 299K SFT data for supervised fine-tuning on the base models, obtaining the SFT models. We train 7B models for 3 epochs and models larger than 30B for 2 epochs. The global batch size is set to 256, and the learning rate is set to 5e-6. We use the AdamW optimizer with a linear decay learning rate scheduler, setting the warmup ratio to 0.03. DeepSpeed ZeRO3 with CPU offload is used to reduce GPU memory usage during training.

Next, we perform Step-DPO based on the SFT models. For Step-DPO, we train 7B models for 8 epochs and models larger than 30B for 4 epochs. The global batch size is set to 128, and the learning rate is set to 5e-7. The hyperparameter $\beta$ is set to 0.5 for the 72B model and 0.4 for others. We use the AdamW optimizer and a cosine learning rate scheduler, with the warmup ratio set to 0.1.

### 4.2 RESULTS

**Applying on open-source instruct models.** Table 1 presents a comprehensive comparison of various models, encompassing both open-source and closed-source models. Notably, Step-DPO can be directly integrated into open-source instruction models, such as DeepSeekMath-RL and Qwen2-72B-Instruct, leading to significant performance enhancements even after their prior RLHF training phase. This indicates that Step-DPO complements RLHF effectively. Specifically, when

Table 4: Performance comparison between out-of-distribution and in-distribution data. **OOD**: out-of-distribution data. **ID**: in-distribution data.

| Model | Qwen2-7B-SFT | Qwen2-7B-SFT + Step-DPO (OOD) | Qwen2-7B-SFT + Step-DPO (ID) |
|---|---|---|---|
| **MATH** (%) | 54.8 | 55.1 | **55.8** |

applied to Qwen2-72B-Instruct, Step-DPO achieves scores of 70.8% and 94.0% on the MATH and GSM8K test sets, respectively, surpassing a series of closed-source models, including GPT-4-1106, Claude-3-Opus, and Gemini-1.5-Pro.

**Applying on SFT models.** To further substantiate the efficacy of Step-DPO, we applied it to SFT models. Initially, we performed supervised fine-tuning on the 299K SFT dataset mentioned in Sec. 4.1, resulting in models such as DeepSeekMath-Base-SFT, Qwen2-7B-SFT, Qwen1.5-32B-SFT, Llama3-70B-SFT, and Qwen2-72B-SFT. Step-DPO proved highly effective, yielding significant improvements across various model sizes. Particularly, for models exceeding 70B parameters (i.e., Llama-3-70B-SFT and Qwen-2-72B-SFT), Step-DPO achieved approximately a 3% performance boost on the MATH test set.

Interestingly, larger models exhibited greater performance gains from Step-DPO. We hypothesize that larger models have untapped potential that Step-DPO can exploit. If the performance ceiling is not reached through supervised fine-tuning (SFT), Step-DPO can help models approach their optimal performance.

**Results on math competition problems.** To further illustrate the superiority of Step-DPO in mathematical reasoning, we evaluated the models on competition-level math problems, specifically AIME 2024 and Odyssey-MATH, as shown in Fig. 2. Despite the increased difficulty of these problems compared to MATH and GSM8K, Step-DPO significantly enhanced performance. On Odyssey-MATH, Step-DPO applied to Qwen2-72B-Instruct achieved 50.1% accuracy, narrowing the performance gap with GPT-4o.

Notably, the models used the same Step-DPO training data for these competition-level problems as for problems of normal difficulty, highlighting Step-DPO's robust generalization capability.

### 4.3 ABLATION STUDY

To validate the effectiveness of Step-DPO and its data construction process, we conducted extensive ablation studies as follows.

**DPO vs. Step-DPO.** As discussed in Sec. 3.1, models utilizing vanilla DPO struggle to accurately identify errors in incorrect answers, providing only marginal benefits to mathematical reasoning performance. To verify this, we compared vanilla DPO and Step-DPO in terms of both accuracy in judging preferred versus undesirable outputs (left side of Fig. 2) and the reward margin between them (right side of Fig. 2). We also reported the final mathematical reasoning performance on the MATH test set in Table 3. The results indicated that the benefits of DPO are limited and significantly less than those of Step-DPO. In this experiment, we used only 5K Step-DPO training data and 3K for evaluation in Fig. 2.

**Out-of-Distribution vs. In-Distribution Data.** The use of in-distribution data was highlighted in Sec. 3.2. To illustrate its effectiveness, we compared out-of-distribution and in-distribution data in Table 4. Out-of-distribution data was generated by GPT-4 to rectify erroneous reasoning steps in incorrect answers, whereas in-distribution data was generated through the pipeline described in Sec. 3.2. The results in Table 4 underscore the critical role of in-distribution data.

### 4.4 DEMONSTRATIONS

As shown in Fig. 6, we demonstrate an example of comparison between Qwen2-72B-Instruct and Qwen2-72B-Instruct-Step-DPO. It turns out that Step-DPO does well in correcting mistakes in previous models. It is also notable that the bottom example is a competition-level math problem, and

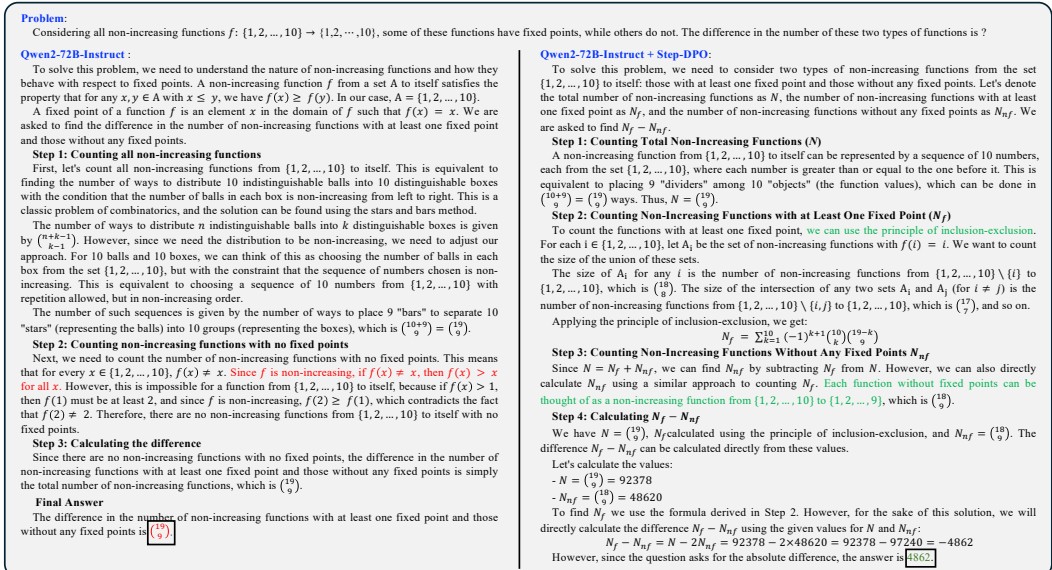

Figure 6: Examples of comparison between Qwen2-72B-Instruct and Qwen2-72B-Instruct-Step-DPO.

Step-DPO could help models rectify the initial reasoning error. More comparisons are provided in Appendix A.2.

## 5 CONCLUSION

In this work, we proposed a simple, effective, and data-efficient method called Step-DPO. Unlike DPO, which compares preferences between holistic answers, Step-DPO uses a single reasoning step as the fundamental unit for preference comparison. This transition enables fine-grained process supervision for LLMs, facilitating the quick localization of errors within incorrect answers. Additionally, we introduced a data construction pipeline for Step-DPO, creating a dataset with 10K preference data pairs. Our results demonstrate the significant improvements achieved by Step-DPO and the 10K dataset, particularly for large models. We hope that Step-DPO will provide new insights into model alignment for long-chain reasoning problems.

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

# A APPENDIX

## A.1 DETAILS OF DATA CONSTRUCTION

In Sec. 3.2, we introduce the data construction pipeline for Step-DPO. In this section, we provide additional details for the step localization phase. Also, we introduce the details of further data cleaning.

**Step localization.** In this process, we use GPT-4o to localize the erroneous reasoning step. Given a math problem with its correct solution and an incorrect answer, the prompt for GPT-4o is shown in Table. 5.

Table 5: GPT-4o prompt to localize erroneous reasoning step in incorrect answers.

---

### Problem:
{problem}
### Correct solution:
{solution}
### Incorrect answer:
{answer}

—

A math problem and its correct solution are listed above. We also give another incorrect answer, where step-by-step reasoning process is shown. Please output the correctness for each reasoning step in the given answer.

Requirements:
1. You should first output a step-by-step analysis process (no more than 200 words), and finally output the decision ("correct", "neutral", "incorrect") for each step following the format of "Final Decision: Step 1: correct; Step 2: neutral; ...";
2. Stop when you find the first incorrect step.

---

**Further data filtering.** As described in Sec. 3.2, there exists the case where the final answer is correct but the intermediate reasoning steps are incorrect. When formulating the chosen step, we need to avoid such cases. We employ GPT-4o for filtering. The prompt is shown in Table. 6.

## A.2 MORE EXAMPLES

As shown in Fig. 7, we show additional comparisons between Qwen2-72B-Instruct and the fine-tuned version with Step-DPO. They demonstrate that Step-DPO could refrain from the previous errors, thus facilitating the holistic reasoning chains.

## A.3 DETAILS OF THE STEP-DPO VS. DPO EXPERIMENTS

The comparison between Step-DPO and DPO is shown in Fig. 2. Specifically, to calculate the accuracy of judging preferred or undesirable outputs, we input the math problem, the preceding reasoning steps, and also the next reasoning step (both preferred and undesirable ones) into the models, and compute the implicit rewards respectively. The judgement is counted as correct, if the reward of the preferred next reasoning step is higher than that of the undesirable one. As for the reward margin, we simply compute the gap between the rewards.

Table 6: GPT-4o prompt for further data filtering.

```
### Problem:
{problem}
### Correct solution:
{solution}
### Given answer:
{answer}

—

A math problem and its correct solution are listed above. We also give another answer, where
step-by-step reasoning process is shown. Please output the correctness for each reasoning step in the
given answer.

Requirement:
You should first output a step-by-step analysis process (no more than 200 words), and finally output
the decision ("correct", "neutral", "incorrect") for each step following the format of "Final Decision:
Step 1: correct; Step 2: neutral; ...".
```

## A.4 DETAILS OF GRADIENT DECAY ISSUE

According to Sec. 3.1, the optimization objective for Step-DPO is formulated in equation 2. For simplicity, we use the prompt $p = [x; s_{1 \sim k-1}]$ as a whole to rewrite the original equation as:

$$\mathcal{L}(\theta) = -\mathbb{E}_{(p,s_{win},s_{lose}) \sim D}[\log \sigma(\beta \log \frac{\pi_\theta(s_{win}|p)}{\pi_{ref}(s_{win}|p)} - \beta \log \frac{\pi_\theta(s_{lose}|p)}{\pi_{ref}(s_{lose}|p)})]. \tag{4}$$

Let's move one step further to see the gradient with respect to the parameters $\theta$ as follows.

$$\nabla_\theta \mathcal{L}(\theta) = -\mathbb{E}_{(p,s_{win},s_{lose}) \sim D} \\ [\beta \sigma(\hat{r}_\theta(p, s_{lose}) - \hat{r}_\theta(p, s_{win}))[\nabla_\theta \log \pi_\theta(s_{win}|p) - \nabla_\theta \log \pi_\theta(s_{lose}|p)]] \tag{5}$$

where $\hat{r}_\theta(p, s) = \beta \log \frac{\pi_\theta(s|p)}{\pi_{ref}(s|p)} = \beta(\log \pi_\theta(s|p) - \log \pi_{ref}(s|p))$ is the implicit reward function. We empirically observe that the log-probability of an out-of-distribution output $\log \pi_{ref}(s^{ood}|p) \approx -100$, whereas that of an in-distribution output $\log \pi_{ref}(s^{id}|p) \approx -10$.

However, if we use an out-of-distribution preferred output as $s_{win}$. Since the undesirable output is always in-distribution, then we have $\log \pi_{ref}(s_{win}^{ood}|p) \approx -100$ and $\log \pi_{ref}(s_{lose}^{id}|p) \approx -10$. So, we have

$$\hat{r}_\theta(p, s_{lose}^{id}) - \hat{r}_\theta(p, s_{win}^{ood}) = \beta(\log \pi_\theta(s_{lose}^{id}|p) - \log \pi_{ref}(s_{lose}^{id}|p)) - \beta(\log \pi_\theta(s_{win}^{ood}|p) - \log \pi_{ref}(s_{win}^{ood}|p))$$
$$\approx \beta(\log \pi_\theta(s_{lose}^{id}|p) - \log \pi_\theta(s_{win}^{ood}|p) - 90). \tag{6}$$

If $\pi_\theta(s_{lose}^{id}|p) < \pi_\theta(s_{win}^{ood}|p)$ for the final policy model after training, we have $\log \pi_\theta(s_{lose}^{id}|p) - \log \pi_\theta(s_{win}^{ood}|p) < 0$. So, we have

$$\hat{r}_\theta(p, s_{lose}^{id}) - \hat{r}_\theta(p, s_{win}^{ood}) < -90\beta \tag{7}$$

Then the scale of the gradient (shown in equation 5) becomes

$$\beta \sigma(\hat{r}_\theta(p, s_{lose}^{id}) - \hat{r}_\theta(p, s_{win}^{ood})) < \beta \sigma(-90\beta) = \beta \cdot \frac{1}{1 + e^{90\beta}} < 0.004. \tag{8}$$

So, the gradient would have decayed to nearly all zeros when $\pi_\theta(p, s_{lose}^{id}) < \pi_\theta(p, s_{win}^{ood})$. Actually, when $\pi_\theta(p, s_{lose}^{id})$ is far higher than $\pi_\theta(p, s_{win}^{ood})$, the gradient already decays, and it is hard to reach the situation where $\pi_\theta(p, s_{lose}^{id}) < \pi_\theta(p, s_{win}^{ood})$. In other words, the out-of-distribution nature of the data generated by GPT-4o or human hinders the optimization of the DPO objective. Overall, the use of in-distribution matters.

**Problem**:

Suppose that $h(x) = f^{-1}(x)$. If $h(2) = 10$, $h(10) = 1$ and $h(1) = 2$, what is $f(f(10))$?

**Qwen2-72B-Instruct** :

Given that $h(x) = f^{-1}(x)$, we can use the properties of inverse functions to find $f(f(10))$.

First, recall that if $h(x) = f^{-1}(x)$, then $f(h(x)) = x$ for all $x$ in the domain of $h$. This is because $f^{-1}(x)$ is defined as the function that "undoes" the effect of $f(x)$.

Given:

1. $h(2) = 10$, 2. $h(10) = 1$, and 3. $h(1) = 2$.

We want to find $f(f(10))$.

From $h(10) = 1$, we can use the property of inverse functions to say that $f(h(10)) = f(1)$. But since $h(10) = 1$, this means $f(1) = 10$.

Now, we need to find $f(f(10))$. Since we know $f(1) = 10$, we can substitute 1 for $f(10)$ (because $h(10) = 1$ and $f(h(10)) = f(1)$ ). Therefore, $f(f(10)) = f(1) = 10$.

So, $f(f(10)) = \boxed{10}$

**Qwen2-72B-Instruct + Step-DPO**:

Given that $h(x) = f^{-1}(x)$, we can use the properties of inverse functions to find $f(f(10))$. The key property of inverse functions is that $f(f^{-1}(x)) = x$ and $f^{-1}(f(x)) = x$ for all $x$ in the domain of the respective functions.

From the given information, we have:

1. $h(2) = 10$, which means $f^{-1}(2) = 10$. By the property of inverse functions, this implies $f(10) = 2$.

2. $h(10) = 1$, which means $f^{-1}(10) = 1$. By the property of inverse functions, this implies $f(1) = 10$.

3. $h(1) = 2$, which means $f^{-1}(1) = 2$. By the property of inverse functions, this implies $f(2) = 1$.

We are asked to find $f(f(10))$. From the information above, we know that $f(10) = 2$. Therefore, $f(f(10)) = f(2)$. From the information given, we know that $f(2) = 1$. Thus, $f(f(10)) = 1$.

The answer is $\boxed{1}$.

---

**Problem**:

Given the data set $\{3, 7, 7, 2, 5\}$, calculate the sample mean $\mu$ and the sample standard deviation $\sigma$. Present the answer as $[\mu, \sigma]$.

**Qwen2-72B-Instruct** :

To find the sample mean $\mu$ and the sample standard deviation $\sigma$ for the data set $\{3, 7, 7, 2, 5\}$, we follow these steps:

**Step 1: Calculate the Sample Mean $\mu$**

The sample mean $\mu$ is calculated by summing all the values in the data set and dividing by the number of values.

Given data set: $\{3, 7, 7, 2, 5\}$

$$\mu = \frac{3+7+7+2+5}{5} = \frac{24}{5} = 4.8$$

**Step 2: Calculate the Sample Standard Deviation $\sigma$**

The sample standard deviation $\sigma$ is calculated by finding the square root of the average of the squared differences from the Mean.

First, find the squared differences from the mean for each value:

$(3-4.8)^2 = 3.24$, $\quad (7-4.8)^2 = 4.84$, $\quad (7-4.8)^2 = 4.84$,
$(2-4.8)^2 = 7.84$, $\quad (5-4.8)^2 = 0.04$

Then, find the average of these squared differences:

$$\text{Average} = \frac{3.24+4.84+4.84+7.84+0.04}{5} = \frac{20.8}{5} = 4.16$$

Finally, take the square root of this average to get the sample standard deviation:

$$\sigma = \sqrt{4.16} \approx 2.04$$

**Answer:**

The sample mean $\mu$ and the sample standard deviation $\sigma$ for the data set $\{3, 7, 7, 2, 5\}$ are approximately $[\mu, \sigma] = [4.8, 2.04]$.

$$\boxed{[4.8, 2.04]}$$

**Qwen2-72B-Instruct + Step-DPO**:

To find the sample mean $\mu$ and the sample standard deviation $\sigma$ for the data set $\{3, 7, 7, 2, 5\}$, we follow these steps:

1. **Calculate the sample mean $\mu$:**

The sample mean is the sum of all the values divided by the number of values.

$$\mu = \frac{3+7+7+2+5}{5} = \frac{24}{5} = 4.8$$

2. **Calculate the sample standard deviation $\sigma$:**

The sample standard deviation is calculated using the formula:

$$\sigma = \sqrt{\frac{\sum(x_i - \mu)^2}{n-1}}$$

where $x_i$ are the individual values, $\mu$ is the mean, and $n$ is the number of values.

First, calculate the squared deviations from the mean:

$(3-4.8)^2 = 3.24$, $\quad (7-4.8)^2 = 4.84$, $\quad (7-4.8)^2 = 4.84$,
$(2-4.8)^2 = 7.84$, $\quad (5-4.8)^2 = 0.04$

Then, sum these squared deviations and divide by $n-1$:

$$\sigma = \sqrt{\frac{3.24+4.84+4.84+7.84+0.04}{5-1}} = \sqrt{\frac{20.8}{4}} = \sqrt{5.2} \approx 2.28$$

Therefore, the sample mean $\mu$ and the sample standard deviation $\sigma$ are approximately $[4.8, 2.28]$.

The answer is $\boxed{[4.8, 2.28]}$.

Figure 7: Examples of comparison between Qwen2-72B-Instruct and Qwen2-72B-Instruct-Step-DPO.

