# OpenReview forum: "Step-DPO: Step-wise Preference Optimization for Long-chain Reasoning of LLMs"
_ICLR.cc/2025/Conference — Submitted to ICLR 2025_

### Official Review · Reviewer_bcGx · 2024-10-30

**Soundness:** 2
**Presentation:** 2
**Contribution:** 1
**Rating:** 1
**Confidence:** 4

**Summary:**

This paper proposes Step-DPO, a new approach aimed at improving the mathematical reasoning abilities of LLMs. The core idea is derived from process supervision [1] and incorporated into DPO.

**Strengths:**

- The method is simple and intuitive.

- The experiments show that Step-DPO can improve the baseline mathematical reasoning ability.

**Weaknesses:**

- The main experiments in Table 1 and Table 2 should include the baseline of naive DPO experiments.

- The author uses different source data for SFT and Step-DPO separately. If we use the whole dataset adopted in SFT and Step-DPO in the SFT phase [2], how would the performance of SFT and DPO compare? This could establish a strong baseline.

- If we construct a paired dataset, the author can use the naive DPO loss function to optimize the LLMs. Therefore, this paper could be regarded as a data construction pipeline for step-level paired data.

- Considering the process supervision proposed in [1] and related works [4], the novelty and data construction methods are limited.

- Accuracy gains are small, many < 1%.

**Questions:**

- I don’t understand why it is “general” after using a math dataset to fine-tune the general models. If the model is specific, the author should report other datasets, for example, in Qwen 2.5-math. If the author claims the models are general, they should be encouraged to discuss the impact of math-specific data fine-tuning [3].

- This paper uses GPT-4 to find the wrong steps. How accurate is GPT-4 at finding the wrong steps?

- Why does the ablation study use only 5k data, while the main experiments use 10k data?



---

[1]. Let’s Verify Step by Step.

[2]. Scaling Relationship on Learning Mathematical Reasoning with Large Language Models.

[3]. https://github.com/QwenLM/Qwen2.5-Math

[4]. Math-Shepherd: Verify and Reinforce LLMs Step-by-step without Human Annotations

---

### Official Review · Reviewer_hykA · 2024-10-30

**Soundness:** 3
**Presentation:** 3
**Contribution:** 2
**Rating:** 5
**Confidence:** 3

**Summary:**

This work studies the improvement of LLM's mathematical reasoning abilities. The authors propose a step-wise DPO method to help LLMs correct errors during the process. The authors construct a step-wise preference dataset through a data construction pipeline, pointing out that using outputs from GPT-4 or humans in the data generation process may lead to out-of-distribution data and ineffectiveness, suggesting the use of a policy model to assist in data generation. The authors conduct experiments on MATH and GSM8K, demonstrating that step-DPO outperformed the original DPO method.

**Strengths:**

1. The mathematical capabilities of Large Language Models (LLMs) have garnered significant attention within the research community, highlighting the importance and relevance of this area of study.
2. The proposal is technically sound, and step-level supervision will undoubtedly bring further improvements to the original trajectory-level DPO.
3. The overall paper is well-presented and easy to follow.

**Weaknesses:**

1. In terms of results, this work ultimately achieved improvements of 1.4% and 1.6% relative to DPO on MATH and GSM8K respectively using Qwen2-72B. My research area is primarily RL, and I'm new to LLMs. Therefore, I won't make judgments on the significance of the performance improvements in the paper. In this regard, I will refer to other reviewers' opinions later. Additionally, I have some questions about the technical design.
2. The construction of the dataset is the core foundation of the proposed method, but the entire annotation and generation process is based on GPT-4 and the initial model $\pi_{ref}$. How is the quality of the data and annotations controlled? I understand that the authors mention relying on models rather than human effort can make this process user-friendly, but it also makes the process a black box and uncontrollable. In this process, it seems we can only rely on comparing performance results after the fine-tuning process to determine the effectiveness of the method.

The current data construction relies on two basic assumptions: (1) There exists a powerful model like GPT-4 that can accurately point out erroneous steps in the reasoning chain. (2) It relies on $\pi_{ref}$ to generate effective step-wise rectification.
These two assumptions are problematic and are not sufficiently discussed in the paper. The experimental results demonstrate that the proposed data pipeline can indeed help the learning of step-wise DPO. However, what are the underlying principles, and what new insights can it provide to researchers in the field? Discussion of these questions can help further enhance the impact of this work.

3. If relying on the existence of a stronger model, or even one that is close to a ground-truth model, then this process is not simply about improving the model's reasoning ability but is closer to knowledge distillation.
4. What kind of improvement can the step-wise rectification based on $\pi_{ref}$ provide? In lines 276-283, the authors mention sampling the referred model multiple times and retaining results where the final answer matches the ground truth. How large is the scope of this data, and does it represent capabilities that the model already possesses but have not been exploited? Moreover, if only mining this part of the capability, the improvement the model can achieve seems to be limited.

**Questions:**

Please refer to the weaknesses 2, 3 and 4 for further clarification.

---

> ### Comment · Reviewer_hykA · 2024-12-02
>
> The authors have not provided a response. My concerns remain.

---

### Official Review · Reviewer_mnZY · 2024-11-04

**Soundness:** 2
**Presentation:** 3
**Contribution:** 2
**Rating:** 3
**Confidence:** 4

**Summary:**

The authors present Step-DPO, a variation on DPO specifically designed to encourage proper reasoning in intermediate reasoning steps, with applications for many-step math reasoning. The method simply uses a differing intermediate reasoning step as the preferred/rejected completion, with the question and all preceding steps as the context. The authors finetune models with this procedure and demonstrate that it leads to improved mathematical reasoning performance.

**Strengths:**

1. A clearly described procedure for curating DPO training data pairs for mathematical reasoning, with a useful step-partitioning procedure.
2. Step-DPO pushes the performance on certain test sets for several SFT models.
3. The paper's diagrams, charts, and communication clearly describe the contribution and results.
4. Dataset creation instructions are useful for those wishing to replicate or extend these results, or generally create good mathematical reasoning models.
5. Analyses about DPO on in vs. out-of-distribution data are helpful insights.

**Weaknesses:**

1. The significance of the new method's dominance over existing models/results is not clear in several cases. The lack of meaningful error bars in figures and tables means that some improvements appear possibly marginal in Tables 1, 3, and 4.
2. In some cases, such as for Qwen-72B-Instruct, the bulk of improvements seems to come from the updated prompting strategy used by the authors, not the Step-DPO method. For example, the abstract concludes with the statement that the Step-DPO-trained Qwen-72B-Instruct surpasses GPT-4-1106, Claude-3-Opus, and Gemini-1.5-Pro on GSM8K and MATH. In fact, it seems it does not surpass Claude-3-Opus on GSM8K. Additionally, Qwen-72B-Instruct already surpassed all of these three models on MATH (and GPT-4-1106 and Gemini-1.5-Pro on GSM8K) after the improved prompting strategy, meaning that the conclusion that Step-DPO allowed for these results seems possibly unfounded.
3. In terms of contribution, it is unclear to me that the framing of Step-DPO as a DPO variant is founded. It seems to me that the formulation in Equation 2 is precisely that of Equation 1, with the exception of changing what is the context and completion. I am not sure why we need a full equation statement of Step-DPO. The paper does mention that "Rejecting an entire undesirable answer in DPO may also discard preceding correct reasoning steps, introducing significant noise and negatively impacting training." (201-203). I agree that there is a major issue with the formulation of DPO for tasks with only some portions that are wrong; it could diminish the likelihood of initial correct reasoning steps. However, as long as the portion of the response before the deviation point is the same, it is effectively part of the context and therefore no gradients will be derived from it. This means that it seems to me that the approach presented in the paper is effectively exactly the same as ordinary DPO, but with the initial portion of both completions identical and the portion of the completion after the deviation clipped after one reasoning step. Mostly, the contribution of this work seems to be pair-preference dataset creation for a targeted purpose.

**Questions:**

1. Are there plans to release the curated dataset for reproducibility? I didn't find any mention of this in the draft, but I may have missed it.
2. Some recent variations on DPO, such as DPO-positive from (Smaug: Fixing Failure Modes of Preference Optimisation with DPO-Positive, Pal et al., 2024), or ORPO (ORPO: Monolithic Preference Optimization without Reference Model, Hong et al., 2024), can mitigate failure modes of DPO, especially the failure mode of decreasing the likelihood of the preferred completion unintentionally. I would be interested to see how {Step-DPO on your treatment of your data} fares against {these methods on a standard DPO treatment of your selected datasets without the stepwise treatment}.

---

> ### Comment · Reviewer_mnZY · 2024-11-29
>
> Note to the authors: only a few more days to communicate with the reviewers wrt this paper.

---

### Official Review · Reviewer_Fsao · 2024-11-07

**Soundness:** 2
**Presentation:** 2
**Contribution:** 2
**Rating:** 5
**Confidence:** 3

**Summary:**

This paper proposes to dpo train the model per each reasoning step. To do so, the authors prepare questions and sampling LLMs to generate reasoning steps, and annotate them to form dpo data triples. Using such data to DPO-train the LLM, it can achieves on-par performance with stoa models on reasoning benchmarks.

**Strengths:**

- The idea of step-wise preference optimization is reasonable - it can guide llm to focus on step-wise correctness.
- The experiments section contains comparison of many models.

**Weaknesses:**

- Key information is not clear: in the so-called step localization stage, about how to label the step-wise reasoning, the authors described "This process can be done manually or using GPT-4". However, it is not clear how this is done in detail. Since this is very important in the entire processing, it would be necessary to know more. Specifically,

   1. how many data is labelled by human, and how many is from GPT-4
   2. For human annotation, how many annotators are hired for each question; what is the question to annotate, e.g., is it a binary classification? Also, since many math questions are not easy, what efforts do the authors take to ensure annotation quality.
   3.  For GPT-4 annotation, what is the prompt? Since model could make mistakes , how to reduce this error risk?
   4.  One alternative (and straightforward) way is to using a small amount of labelled data to fine-tune a llm to be a PRM, and use it to give step-wise supervision siginals.

- In table3, compared to normal DPO, the step-wise DPO's gain is very limited. It's not very convincing to claim the advantage of step-wise  DPO. Also, why only using 5K data for this ablation instead of the entire dataset?
- Similarly in table4, the marginal improvement of in distribution data is hard to conclude it is better.

**Questions:**

Please refer to my questions in the weakness section.

---

### Meta-Review · Area_Chair_b5XP · 2024-12-22

**Metareview:**

This paper presents Step-DPO, which is a variant of DPO that use intermediate reasoning steps as the preference feedback to the model.

The paper is clear, results show that Step-DPO improves performance, and its ablation analyses on DPO vs Step-DPO and in vs out of distribution data show interesting insights.

However, as pointed out by the reviewers, the empirical improvement is relatively small, the baselines are relatively weak (though stronger ones can be constructed), and there are some inconsistency in experiment settings.

The authors did not provide any rebuttal and the concerns from reviewers are not addressed.

Given these reasons, I recommend rejection.

**Additional Comments On Reviewer Discussion:**

The authors did not provide any rebuttal and the concerns from reviewers are not addressed.

---

### Decision · Program_Chairs · 2025-01-22

Reject